# A specific EMC subunit supports Dengue virus infection by promoting virus membrane fusion essential for cytosolic genome delivery

Parikshit Bagchi[1¤‡], Kaitlyn Speckhart[1,2‡], Andrew Kennedy[3], Andrew W. Tai[3,4], Billy Tsai [1,2]*

**1** Department of Cell and Developmental Biology, University of Michigan Medical School, Ann Arbor, Michigan, United States of America, **2** Cellular and Molecular Biology Program, University of Michigan Medical School, Ann Arbor, Michigan, United States of America, **3** Department of Internal Medicine, University of Michigan Medical School, Ann Arbor, Michigan, United States of America, **4** Department of Microbiology and Immunology, University of Michigan Medical School, Ann Arbor, Michigan, United States of America

¤ Current address: Vir Biotechnology, Inc., St. Louis, Missouri, United States of America
‡ These authors are joint senior authors on this work.
* btsai@umich.edu

**Data Availability Statement:** All relevant data are within the manuscript and its Supporting Information files.

## Abstract

Dengue virus (DENV) represents the most common human arboviral infection, yet its cellular entry mechanism remains unclear. The multi-subunit endoplasmic reticulum membrane complex (EMC) supports DENV infection, in part, by assisting the biosynthesis of viral proteins critical for downstream replication steps. Intriguingly, the EMC has also been shown to act at an earlier step prior to viral protein biogenesis, although this event is not well-defined. Here we demonstrate that the EMC subunit EMC4 promotes fusion of the DENV and endosomal membranes during entry, enabling delivery of the viral genome into the cytosol which is then targeted to the ER for viral protein biosynthesis. We also found that EMC4 mediates ER-to-endosome transfer of phosphatidylserine, a phospholipid whose presence in the endosome facilitates DENV-endosomal membrane fusion. These findings clarify the EMC-dependent DENV early entry step, suggesting a mechanism by which an ER-localized host factor can regulate viral fusion at the endosome.

## Author summary

Although DENV infection causes significant morbidity and mortality, no effective DENV antivirals exist. Here we discover a host factor that promotes an important DENV entry step. Specifically, we found that a component of an endoplasmic reticulum protein complex called EMC4 supports fusion of the DENV and endosomal membranes—this event delivers the viral genome into the host cytosol to drive virus infection. By elucidating a crucial DENV entry step, this work may help to develop critically-needed antivirals against DENV infection.

**Funding:** This work was supported by NIH R01GM139823 (B.T., A.T.) and start-up funds from the University of Michigan to B.T. K.S. was supported by NIH T-32-GM007315. The funders had no role in study design, data collection and analysis, decision to publish, or preparation of the manuscript.

**Competing interests:** The authors have declared that no competing interests exist.

## Introduction

As a member of the flavivirus family, Dengue virus (DENV) represents the most common human arboviral infection globally, causing approximately 390 million acute infections, 500,000 hospitalizations, and 25,000 deaths annually [1]. Despite this, the cellular infection mechanism of DENV remains unclear. To infect cells, DENV undergoes receptor-mediated endocytosis to reach the late endosome (LE) [2]. Here, via a low pH-driven process, the viral membrane fuses with the LE membrane–this step is critical because it enables release of the positive-strand (+) RNA genome into the cytosol. These represent the early DENV entry steps.

Upon reaching the cytosol, the RNA genome is targeted to the cytosolic surface of the endoplasmic reticulum (ER) membrane, where the nascent polypeptide chain undergoes co-translational translocation leading to biosynthesis of a single polyprotein encoding three structural (capsid, prM, E) and seven non-structural (NS1, NS2A, NS2B, NS3, NS4A, NS4B, and NS5) proteins. Ensuing polyprotein cleavage by viral and host proteases generates the individual proteins [2,3]. In concert with host components, the non-structural viral proteins exert critical roles in promoting virus replication and assembly. Once assembled, the new viral progeny exits the host cell to complete the infection cycle.

To identify host factors that facilitate flavivirus (including DENV) infection, several groups used the genetic CRISPR/Cas9 loss-of-function approach and found that the 10-subunit ER membrane complex (EMC) supports DENV, and other flavivirus, infection [4–8]. Mechanistically, EMC was shown to act as a transmembrane chaperone, promoting biosynthesis of the NS4A and NS4B viral proteins at the ER membrane [7,9,10]. Because the CRISPR/Cas9 knockout (KO) strategy used in these studies likely depleted the entire EMC, it remains unclear whether a select EMC subunit is sufficient to act as the transmembrane chaperone, or if the entire protein complex is required for this function.

In parallel, reports also raised the possibility that the EMC might act at an earlier step during DENV entry, prior to ER-dependent viral protein biogenesis [11]. Indeed, the EMC was suggested to play a role during successful release of the Zika viral genome into the cytosol [11]. However, how the EMC functions in this early virus entry step—independent of its ER-dependent chaperone activity—is completely unknown. Moreover, the specific EMC subunit that participates in this early DENV entry step is also not defined.

Here, using the siRNA-mediated knockdown (KD) approach to acutely and selectively deplete each of the 10 EMC subunits, our results revealed that the EMC4 subunit of the EMC promotes fusion of the DENV and endosome membranes during entry, enabling delivery of the viral genome into the cytosol. We further demonstrate that EMC4 supports the ER-to-endosome transfer of phosphatidylserine (PS), a phospholipid whose presence in the endosomal membrane facilitates fusion of the DENV and endosome membranes [12]. Our results thus elucidate the early DENV entry step mediated by the EMC, identify the select EMC subunit responsible for this action, and explain how an ER-resident host component can impact a virus fusion event at the late endosome.

## Results

### EMC4 promotes DENV infection

Using a reporter DENV expressing a luciferase reporter gene to assess DENV infection, we and others previously found that DENV infection was blocked under CRISPR-mediated EMC KO [10]. To identify which specific EMC subunit is responsible for this, we used the siRNA approach to individually KD each of the 10 EMC subunits in HEK 293T cells (S1 Fig). Our results showed that although acute KD of some EMC subunits modestly impaired DENV

infection, KD of EMC4 resulted in the most severe phenotype (Fig 1A). For this reason, we focused this study on EMC4. Although EMC5 and EMC6 are thought to be core EMC subunits [13], we found that KD of either of these subunits for 48 hr did not affect the level of EMC4 (S1 Fig). It should be noted that a previous report found that KD of EMC4 did not affect the level of the other EMC subunits [13]. Consistent with the results using HEK 293T cells, DENV infection in the Huh 7.5.1 hepatoma cells were similarly impaired (Fig 1B) when EMC4 was

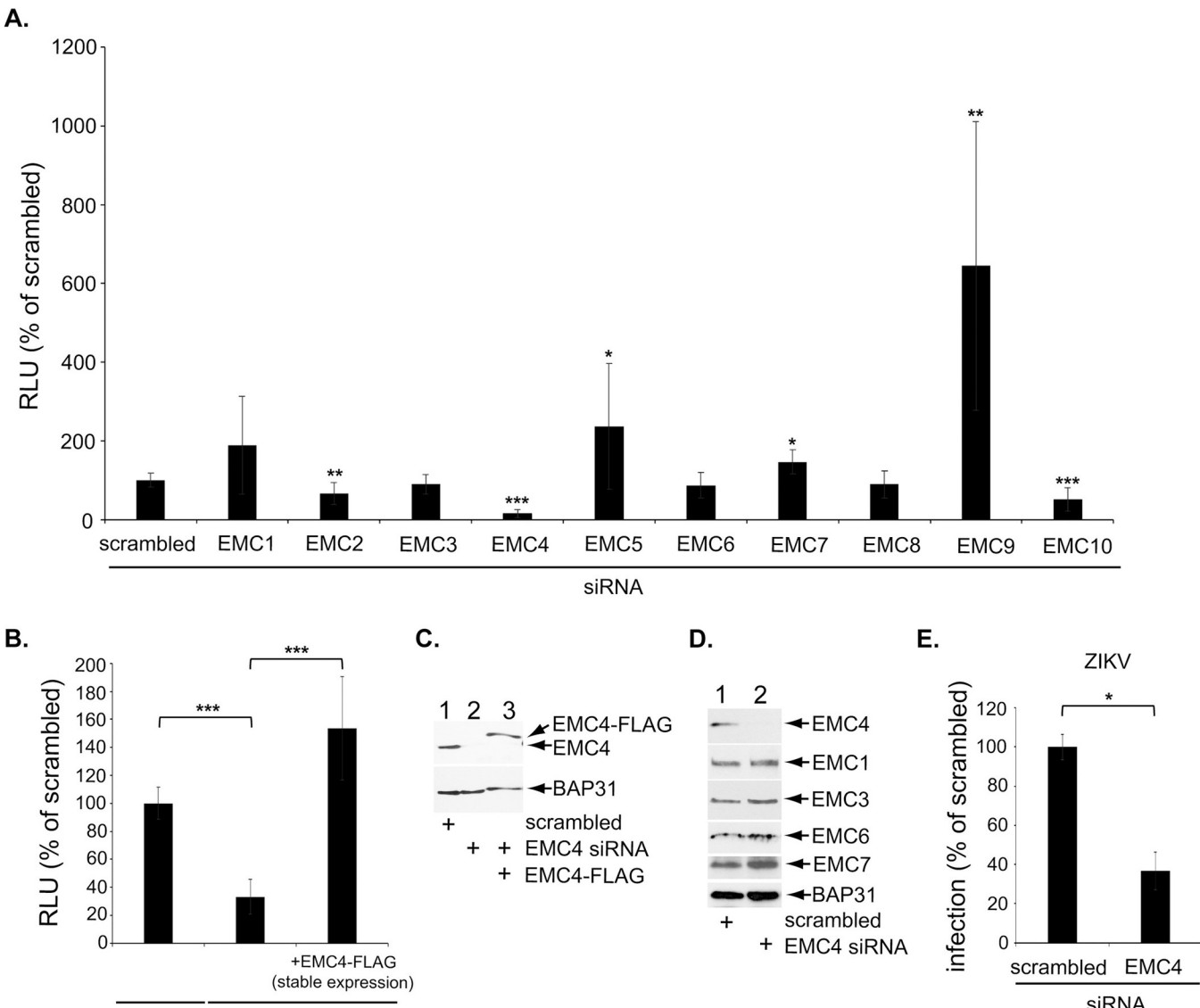

**Fig 1. EMC4 promotes DENV infection. A.** HEK 293T cells were transfected with the indicated siRNAs followed by luc-DENV2 infection (MOI 0.05) for 48 h. The data show relative luciferase unit normalized to the scrambled siRNA, and represent means and standard deviations (SD) ($n \geq 3$). All indicated P values are in comparison to the scrambled condition. **B.** Huh 7.5.1 (or EMC4-FLAG-stably expressing Huh 7.5.1) cells were transfected with the indicated siRNAs followed by luc-DENV2 infection (MOI 0.05) for 48 h. The data show relative luciferase unit normalized to the scrambled siRNA, and represent means and standard deviations (SD) ($n \geq 3$). **C.** Huh 7.5.1 (or EMC4-FLAG-stably expressing Huh 7.5.1) cells were transfected with the indicated siRNAs and lysed with 1% Triton X-100, and the resulting extract was subjected to SDS-PAGE and immunoblotted with the indicated antibodies. **D.** Huh 7.5.1 cells were transfected with the indicated siRNAs and lysed with 1% Triton X-100, and the resulting extract was subjected to SDS-PAGE and immunoblotted with the indicated antibodies. **E.** Huh 7.5.1 cells were transfected with indicated siRNA followed by infection with ZIKV at 0.1 MOI. Supernatants were used to infect naïve Huh 7.5.1 cells and incubated for 3 days before being probed for flavivirus E-protein to determine viral titer. *$P \leq 0.05$, **$P \leq 0.01$, ***$P \leq 0.001$.

depleted (Fig 1C, top panel, lane 2 vs 1). Expression of siRNA resistant EMC4-FLAG in the EMC4-depleted cells (Fig 1C, top panel, lane 3) completely rescued DENV infection (Fig 1B), demonstrating that the block in DENV infection resulting from the EMC4 siRNA is due to loss of EMC4 and not to unintended off-target effects. Importantly, EMC4 KD selectively depleted EMC4 without affecting other EMC subunits at 48 hr post-transfection (Fig 1D). EMC4 KD also decreased infection of the related Zika virus (ZIKV) (Fig 1E). Together these results demonstrate that EMC4 plays a critical role in DENV (and likely other flavivirus) infection.

## EMC4 does not promote ER-dependent biosynthesis of DENV non-structural proteins or viral replication

To determine which DENV entry step is mediated by EMC4, we first used a "mini-gene" plasmid encoding NS4B - a viral protein whose expression is blocked under EMC KO–and unexpectedly found that acute EMC4 KD did not affect NS4B expression (Fig 2A, top panel, lane 2 vs 1). To further support this finding, we used the subgenomic DENV replicon (DENV-2 replicon) system. In this system, cells stably expressing the DENV-2 replicon can express and deliver the viral RNA into the host cytosol without undergoing the typical DENV early entry steps: receptor-mediated endocytosis of DENV to the LE followed by virus-LE membrane fusion to enable genome escape to the cytosol. Hence, this replicon system is useful in identifying viral and host components that support DENV infection after the viral RNA reaches the cytosol.

Using Huh 7.5.1 cells stably expressing the DENV-2 replicon, we found that ER-dependent expression of NS4B is also unaffected by acute EMC4 KD (Fig 2B, top panel, lane 2 vs 1), consistent with the plasmid-based mini-gene expression system (Fig 2A). Additionally, replication of DENV-2 RNA [as monitored by an antibody recognizing the double-stranded RNA (dsRNA) replication intermediate] is not affected under EMC4 KD (Fig 2C, compare top and bottom rows; quantified in Fig 2D), whereas addition of the established RNA polymerase inhibitor NITD008 [14] did block appearance of the dsRNA signal (Fig 2E), as expected. Further, using this replicon system, the viral RNA level in the cell (as assessed by qPCR) is unchanged by loss of EMC4 (Fig 2F). Together these data demonstrate that EMC4 is not, by itself, required for ER-dependent biosynthesis of a viral non-structural protein or viral replication, raising the possibility that EMC4 might act at an early step during DENV entry.

## EMC4 supports cytosol escape of the DENV genome

To test the idea that EMC4 functions at an early DENV entry step, we found that the level of dsRNA staining (Fig 3A, compare top and bottom rows; quantified in Fig 3B) and viral RNA (Fig 3C) decreased dramatically when EMC4 KD cells were infected by the DENV reporter virus (for 48 h), in striking contrast to the replicon system where depletion of EMC4 did not reduce the level of dsRNA staining (Fig 2C; quantified in Fig 2D) or viral RNA (Fig 2E). These data are consistent with the hypothesis that EMC4 acts at an early step during DENV infection which can be bypassed by the replicon system.

These findings therefore prompted us to probe whether EMC4 promotes DENV genome escape into the cytosol from the LE. To test this, we developed a biochemical fractionation approach (see Materials and Methods) to isolate the cytosol (and endosome) fractions (Fig 3D). We found that the cytosol marker actin was present in the cytosol fraction, the late endosome-associated protein Rab7 (as well as the late endosome/lysosome protein LAMP1) in the endosome fraction, and the ER marker BAP31 in neither fraction but instead in the whole cell lysate (WCL). These findings verify the integrity of the fractionation method.

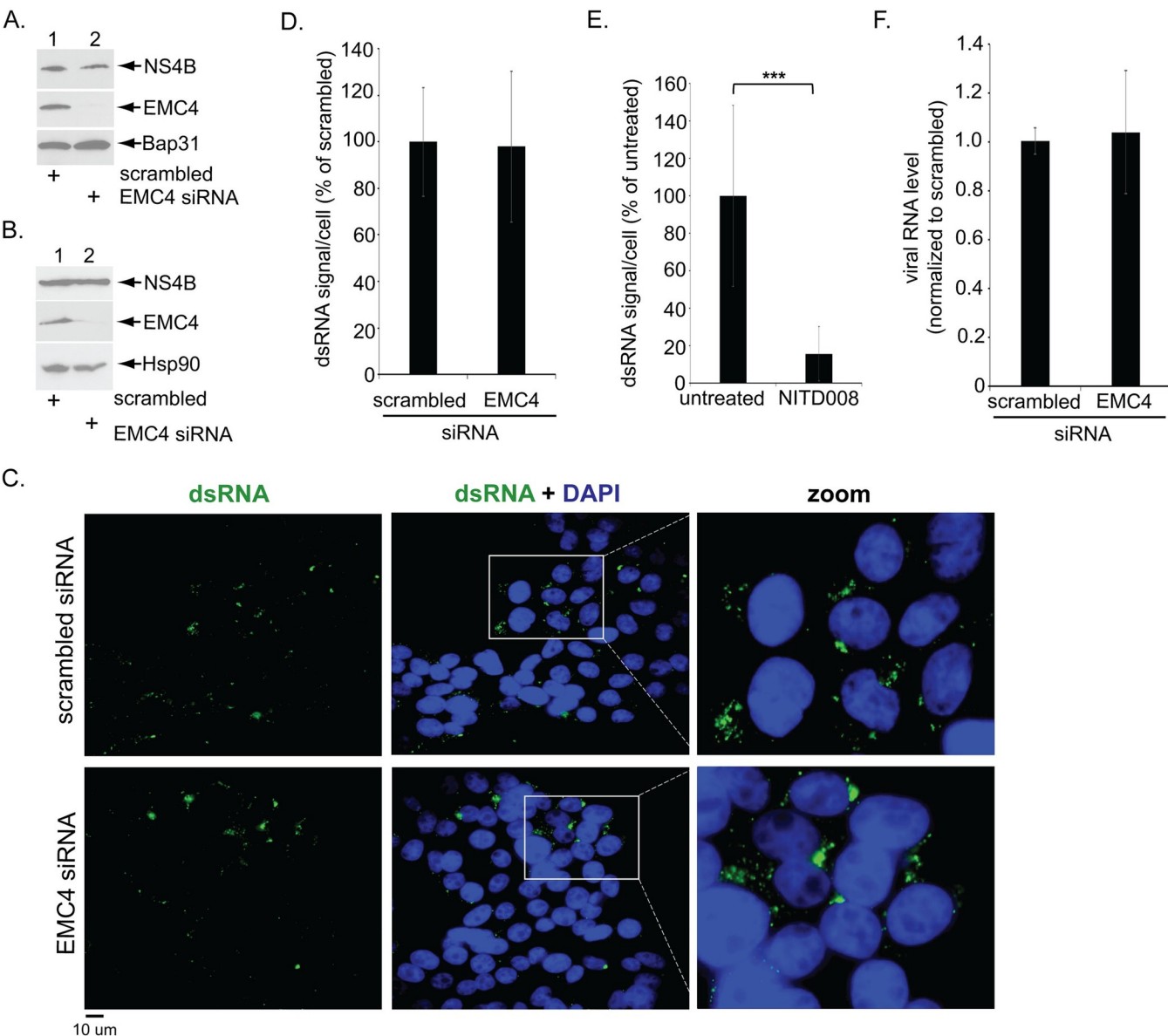

**Fig 2. EMC4 does not promote ER-dependent biosynthesis of a DENV non-structural protein or viral replication. A.** HEK 293T cells were transfected with the indicated siRNAs followed by transfection with the DENV NS4B mini-gene expressing plasmid. The cells were then lysed with 1% Triton X-100, and the resulting extract was subjected to SDS-PAGE and immunoblotting with the indicated antibodies. **B.** Huh 7.5.1 cells stably expressing DENV2 replicon were transfected with the indicated siRNAs and lysed with 1% Triton X-100, and the resulting extract was subjected to SDS-PAGE and immunoblotted with the indicated antibodies. **C.** Huh 7.5.1 cells stably expressing the DENV2 replicon were transfected with the indicated siRNAs, fixed, stained with the indicated antibody, and imaged under widefield epifluorescence microscope. Zoomed-in images of the white box regions are also shown. **D.** Quantification of dsRNA signal/cell from C. **E.** As in D, except cells were treated with or without 10uM NITD008. **F.** Huh 7.5.1 cells stably expressing the DENV2 replicon were transfected with the indicated siRNAs and the total RNA were extracted followed by RT-qPCR to monitor the viral RNA level (normalized to scrambled). ***P ≤ 0.001.

Using this approach, our results showed that—via qPCR—the viral RNA level in the cytosol fraction (derived from cells infected with DENV for 1 h to restrict our analysis to the input virus genome) was significantly decreased under EMC4 KD (Fig 3E, second bar), similar to the effect of treating control cells with chloroquine which blocks low pH-dependent fusion of DENV and endosomal membranes required for genome escape into the cytosol (Fig 3E, third

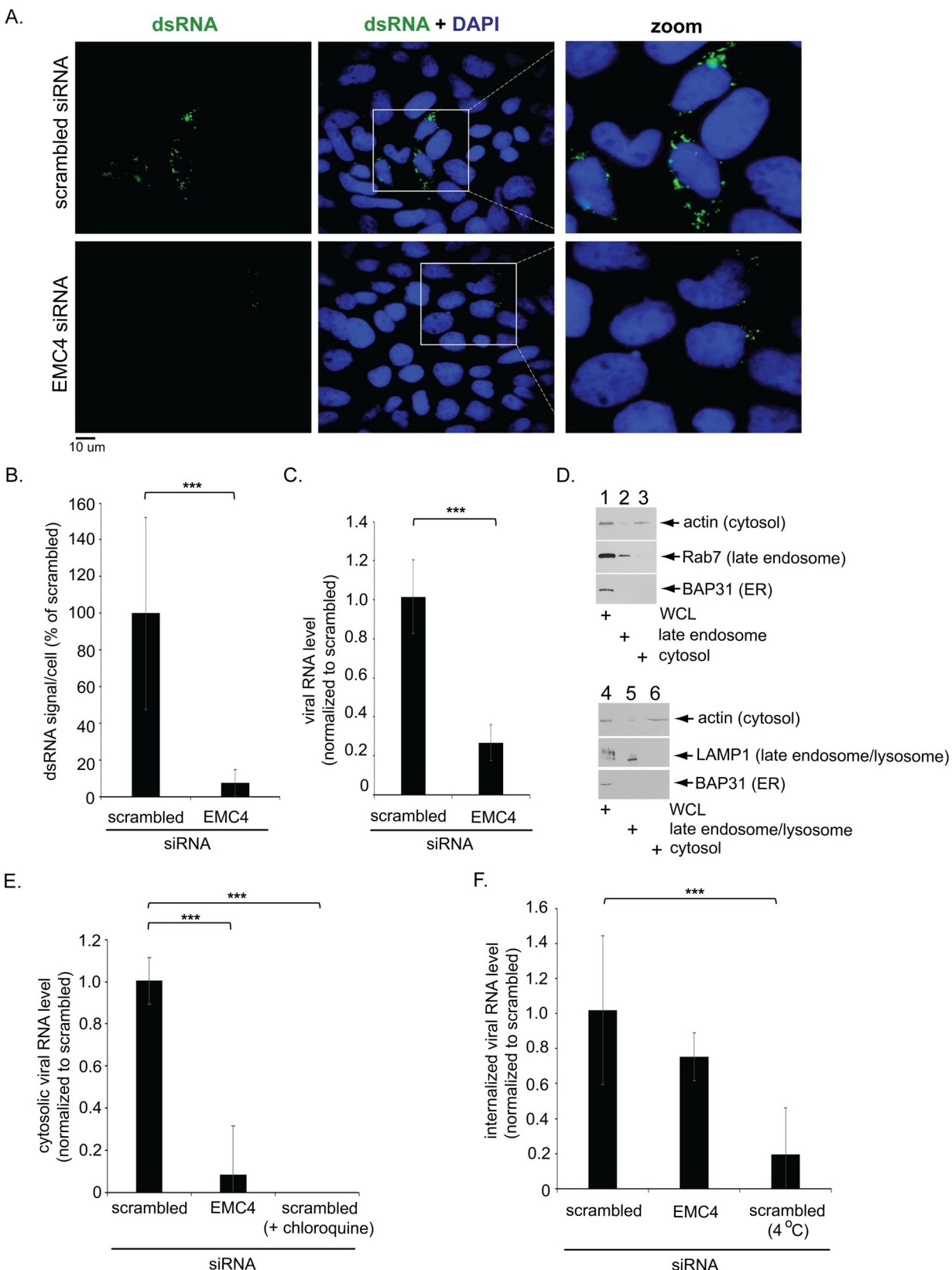

**Fig 3. EMC4 supports cytosol escape of the DENV genome. A.** Huh 7.5.1 cells were transfected with the indicated siRNAs followed by infection with luc-DENV2 (MOI 0.1) for 48 h, fixed, stained with the indicated antibody, and imaged under widefield epifluorescence microscope. Zoomed-in images of the white box regions are also shown. **B.** Quantification of dsRNA signal/cell from A. **C.** Huh 7.5.1 cells were transfected with the indicated siRNAs followed by infection with luc-DENV2 (MOI 0.05) for 48 h and the total RNA were extracted followed by RT-qPCR to monitor viral RNA level (normalized to scrambled). **D.** Whole cell lysate (WCL), endosomal, and cytosolic fractions derived from Huh 7.5.1 cells were subjected to SDS-PAGE and immunoblotted with indicated antibodies. 7% of WCL and 8% of each the endosomal and cytosolic fractions was loaded for western. **E.** Huh 7.5.1 cells were transfected with the indicated siRNAs followed by infection with luc-DENV2 for 1 h (MOI 6) with or without chloroquine and the RNA was extracted from the cytosolic fraction followed by RT-qPCR to determine viral RNA level (normalized to scrambled). The data values representing means and standard deviations (SD) ($n \geq 3$). **F.** Huh 7.5.1 cells were transfected with the indicated siRNAs followed by infection with luc-DENV2 for 1 h (MOI 6) at either 37˚C or 4˚C, treated with Proteinase K for 45 min in ice and the RNA was extracted followed by RT-qPCR to determine viral RNA level (normalized to scrambled). The data values representing means and standard deviations (SD) ($n \geq 3$). ***$P \leq 0.001$.

bar). As a control, the level of internalized virus (at 1 hpi) was not significantly affected by depletion of EMC4 (Fig 3F, second bar), in contrast to control cells incubated at 4˚C which blocks viral internalization (Fig 3F, third bar). Together, these results suggest that EMC4 supports DENV genome release into the cytosol from the LE.

## DENV membrane fusion is dependent on EMC4

Because viral genome release into the cytosol occurs after DENV fuses with the LE membrane, we asked if this membrane fusion event is EMC4-dependent. To assess this, we used an established protocol in which the DENV membrane is labelled with the lipophilic dye [1,1-Dioctadecyl-3,3,3,3-tetramethylindodicarbocyanine] (DiD). The principle of this strategy is straightforward. When DiD is incorporated into the viral membrane at high concentrations, a fluorescence signal is undetected due to self-quenching. In contrast, when the DiD-labelled virus fuses with the LE membrane, DiD diffuses into the host membrane and dequenches, thereby emitting fluorescence. Hence, fluorescence detection from a DiD-labelled DENV monitors the viral-LE membrane fusion event.

Using this method, we found that whereas fluorescence can be readily observed in control cells infected with DiD-labelled DENV (1 hpi), the fluorescence signal decreased robustly under EMC4 KD (Fig 4A, compare top to middle rows; quantified in Fig 4B). As expected, addition of chloroquine to control cells also blocked appearance of the fluorescence signal (Fig 4A, bottom row; quantified in Fig 4B). By contrast, acute KD of EMC3 (Fig 4C)–an EMC subunit that does not support DENV infection (Fig 1A)—did not impair the fluorescence signal (Fig 4D; quantified in Fig 4E). We conclude that DENV membrane fusion is dependent on EMC4.

## EMC4 plays a role in efflux of phosphatidylserine to the endosome

How might the ER-localized EMC4 facilitate DENV membrane fusion at the endosome? EMC4 was previously shown to act as a molecular tether, stabilizing the membrane contact site (MCS) between the ER and endosome [15]. Intriguingly, this MCS was reported to support efflux of the phospholipid PS from the ER to the endosome [16]. Because addition of PS to cells stimulated DENV-LE membrane fusion and virus infection [12], we hypothesized that EMC4-dependent stabilization of the LE-ER MCS mediates the efficient efflux of PS from the ER to the LE.

To test this idea, we measured the endogenous PS level in the LE under EMC4 KD using the fluorescent PS sensor LactC2-GFP [17]. Specifically, we used super-resolution confocal microscopy (structured illumination microscopy, SIM) to visualize the extent of colocalization between LactC2-GFP and an LE marker (STARD3-FLAG). Our results revealed that EMC4 depletion decreased colocalization between LactC2-GFP and STARD3-FLAG (Fig 5A,

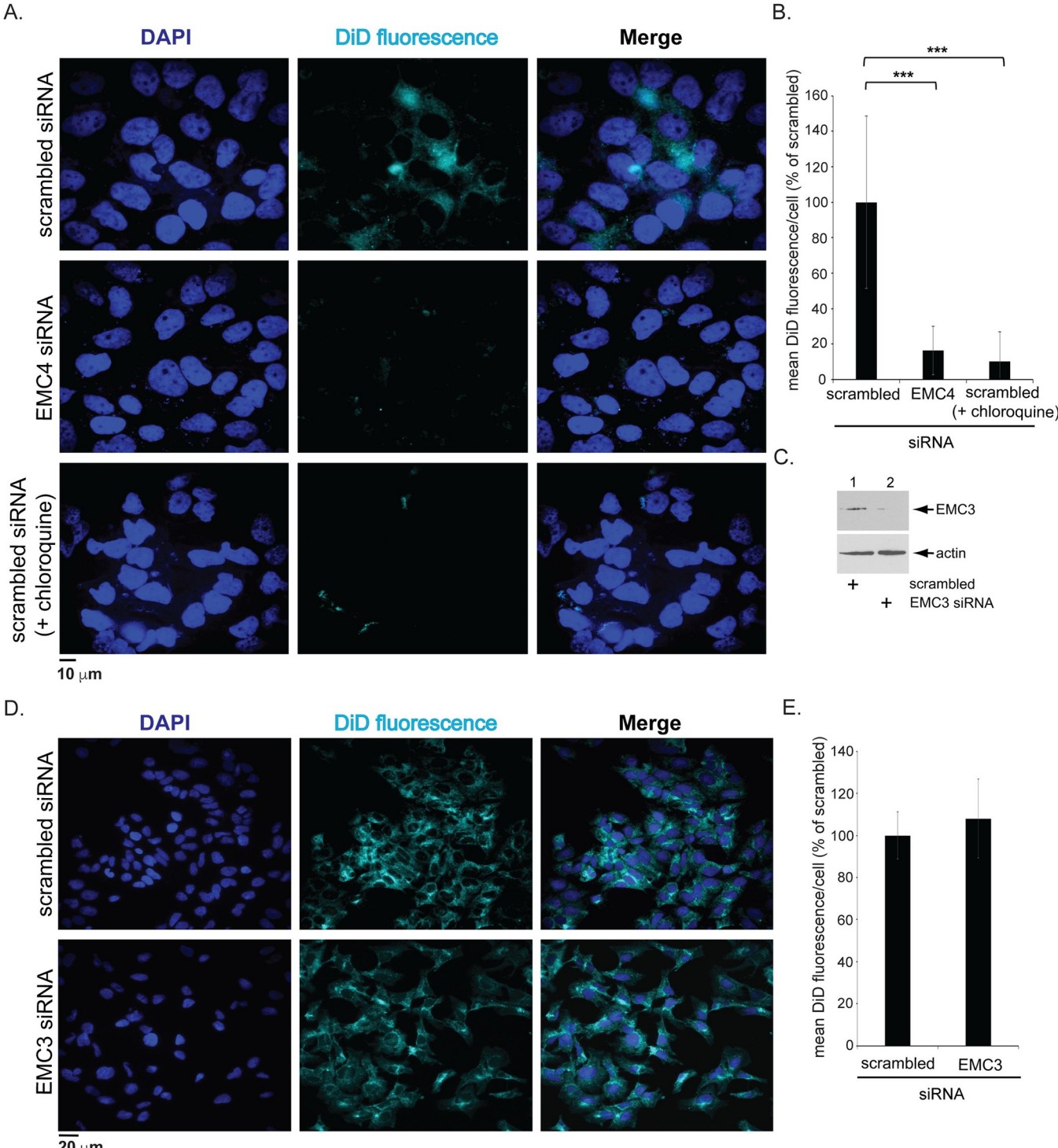

**Fig 4. DENV membrane fusion is dependent on EMC4. A.** Huh 7.5.1 cells were transfected with the indicated siRNAs followed by infection with DiD-labeled luc-DENV2 for 1 h (MOI 5) with or without chloroquine, fixed, and imaged under widefield epifluorescence microscope to determine the DiD and DAPI signals. **B.** Quantification of the mean DiD fluorescence/cell in A (normalized to scrambled). The data values represent the means and standard deviations (SD) ($n \geq 3$). **C.** Extracts from Huh 7.5.1 cells transfected with the indicated siRNAs were subjected to SDS-PAGE followed by immunoblotting with the indicated antibodies. **D.** As in A, except EMC3 KD cells were used. **E.** As in B, except EMC3 KD cells were used for the quantification. ***P $\leq$ 0.001.

compare middle to top row, far right panel; the extent of colocalization was quantified in Fig 5B). As a control, we knocked down EMC3 and found that loss of this EMC subunit did not decrease colocalization between LactC2-GFP and STARD3-FLAG (Fig 5A, bottom row, far right panel; quantified in Fig 5B). These findings suggest that EMC4 plays a role in maintaining the PS level in the LE, which is critical for DENV membrane fusion.

We reasoned that if the block in DENV infection under EMC4 KD is due to a decreased level of endosome-localized PS, supplementing EMC4 KD cells with purified PS in the media–which should spontaneously insert into the plasma membrane, undergo endocytosis to reach the endosome, and bypass the EMC4-dependent ER-to-endosome route–might restore virus infection. To ensure that DENV and PS are present in the same endocytic vesicle during entry, we used a synchronized method. Specifically, DENV and PS were added to cells simultaneously at 4˚C and allowed to enter together upon shifting to 37˚C. Using this approach, DENV infection decreased under EMC4 KD as expected (Fig 5C). Importantly, addition of PS to EMC4 KD cells largely restored the block in DENV infection due to depletion of EMC4 (Fig 5C), suggesting that a lowered PS level in the endosome largely contributes to the attenuation of DENV infection in EMC4-depleted cells. These findings prompted us to test whether biosynthesis of PS affects DENV infection. When the two enzymes responsible for PS synthesis, phosphatidylserine synthase 1 (PTDSS1) and phosphatidylserine synthase 2 (PTDSS2), are simultaneously depleted in cells (Fig 5D, top and bottom panels), DENV infection was impaired (Fig 5E), furthering the idea that PS plays an important role during DENV entry.

Because EMC4 can stabilize the LE-ER MCS by binding to the LE-associated protein Rab7 [15], we asked if Rab7 might play a role during DENV infection. However, in Rab7 KD cells (S2A Fig), DENV infection was unaffected when compared to control cells, in contrast to EMC4 KD cells (S2B Fig). Consistent with this, expression of dominant-negative Rab7 (Rab7 N125I) also did not block virus infection (S2C Fig). Together, these findings suggest that EMC4 uses a Rab7-independent mechanism to promote DENV entry.

## Discussion

During early entry, DENV fuses with the LE membrane to release its RNA genome into the cytosol (Fig 5F). The cytosol-localized viral RNA is then targeted to the ER where the nascent polypeptide chain is co-translationally translocated to produce a single polyprotein which is cleaved to generate viral non-structural and structural proteins. Some of the non-structural proteins in turn assist in viral replication [2,3]. After the replicated genome is assembled with the structural proteins, the newly formed progeny exits the host cell. Via the CRISPR/Cas9 KO strategy, the EMC was reported to play an important role in ER-dependent biosynthesis of DENV polyproteins [7,9,10], although this ER-localized membrane protein complex was also proposed to function at an earlier step prior to viral protein translation [11]. In the case of ZIKV, the EMC was shown to promote cytosol delivery of the viral genome via an unidentified upstream event [11]. The specific EMC-dependent early DENV entry step, the identification of the select EMC subunit that facilitates this step, and the molecular mechanism by which the EMC promotes this event, are unclear. This manuscript addresses these enigmas.

Using the siRNA-mediated KD approach to acutely deplete select EMC subunits, our results reveal that the EMC4 subunit of the EMC supports DENV membrane fusion which is essential for delivery of the viral RNA genome into the cytosol from the LE. Whether a separate pool of EMC4 operates independently of the entire EMC to carry out this function, or if EMC4 exerts this role in the context of the entire EMC, remains unknown. Nonetheless, there is precedent for an individual EMC subunit carrying out a distinct function–isolated EMC1 was shown to be fully capable of acting as a transmembrane chaperone during ER-to-cytosol

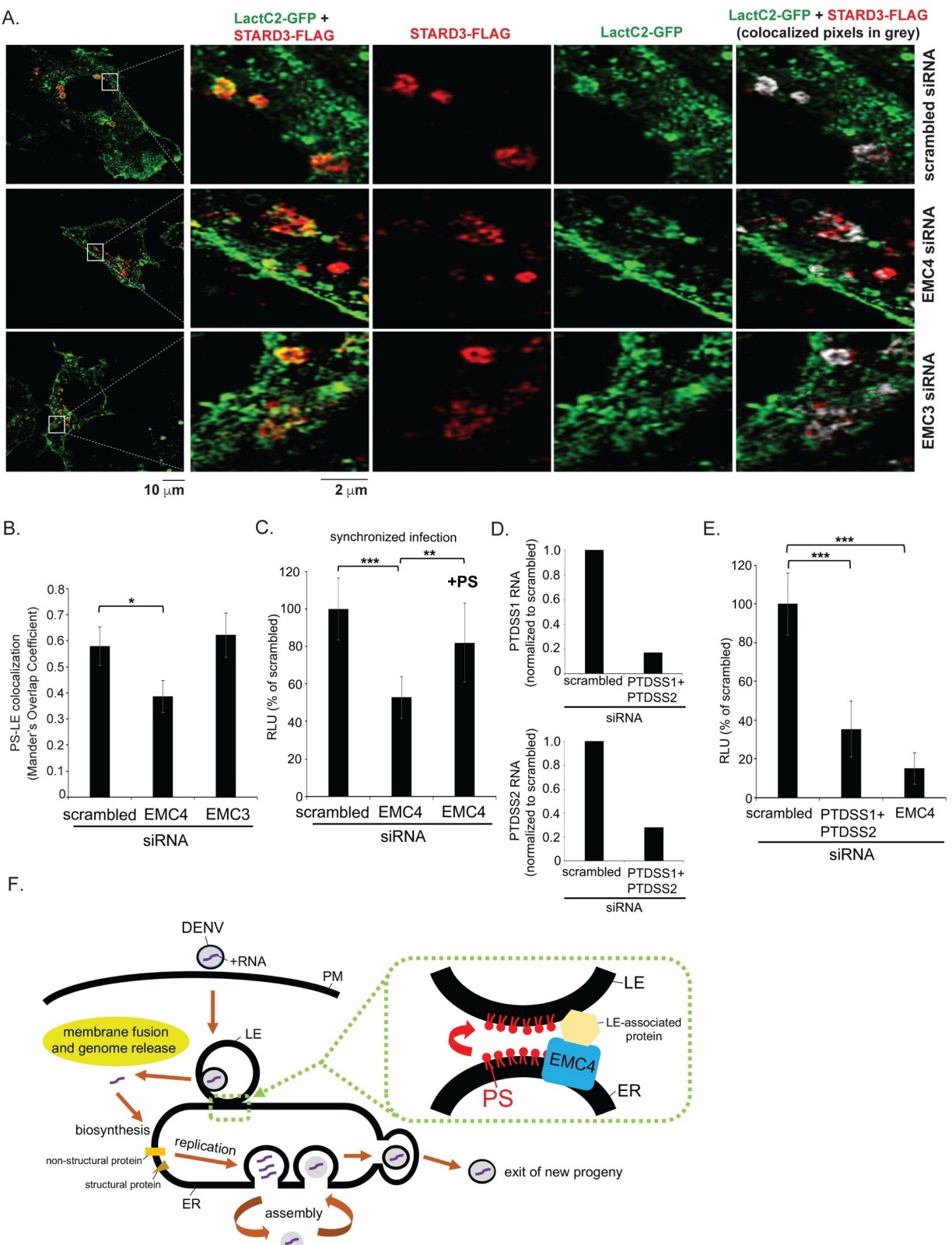

**Fig 5. EMC4 plays a role in efflux of phosphatidylserine to the endosome. A.** Huh 7.5.1 cells transiently expressing LactC2-GFP and STARD3-FLAG were transfected with the indicated siRNAs, fixed, stained with indicated antibodies and imaged by structured illumination

microscopy (SIM). **B.** Quantification of the PS-LE co-localization in A. **C.** Huh 7.5.1 cells were transfected with the indicated siRNAs followed by infection with luc-DENV2 (MOI 3, synchronized) for 48 h with or without 2.6 uM phosphatidylserine (PS). The data show the relative luciferase unit normalized to the scrambled siRNA, and represent means and standard deviations (SD) ($n \geq 3$). **D.** RNA was extracted from Huh 7.5.1 cells depleted of both PTDSS1 and PTDSS2. and subjected to RT-qPCR analysis using the appropriate primers indicated in Table 1. The data are normalized against the scrambled control. **E.** Huh 7.5.1 cells were transfected with the indicated siRNAs followed by infection with luc-DENV2 (MOI 0.05) for 48 h. The data show the relative luciferase unit normalized to the scrambled siRNA, and represent means and standard deviations (SD) ($n \geq 3$). **F.** Model depicting how EMC4 promotes DENV infection. During entry, DENV fuses with the LE membrane to release its RNA genome into the cytosol. The cytosol-localized viral RNA is targeted the ER where the nascent polypeptide chain is co-translationally translocated on the ER membrane to produce a single polyprotein which is cleaved to generate viral non-structural and structural proteins. Some of the non-structural proteins facilitate viral replication. After assembly of the replicated genome with the structural proteins, the newly-formed progeny exits the host cell. EMC4 is proposed to bind to a LE-associated protein to stabilize the LE-ER MCS, enabling efflux of phosphatidylserine (and potentially other lipids) from the ER to the LE. This step facilitates DENV fusion with the LE membrane. $^{*}P \leq 0.05$, $^{**}P \leq 0.01$, $^{***}P \leq 0.001$.

membrane escape of the SV40 polyomavirus [18]. In our study, we did not find that KD of EMC5 or EMC6 affected the level of EMC4 at 48 hr post-transfection, in contrast to a previous report that used a 72 hr duration of knockdown [13]. This could be due to differences in cell type or duration of knockdown.

Mechanistically, we further demonstrated that the PS level in the LE decreased under EMC4 depletion, suggesting that EMC4 facilitates the ER-to-LE transfer of PS. Whether EMC4 assists in the ER-to-LE transfer of other lipids to promote DENV membrane fusion remains unclear. Because a previous report showed that supplementing cells with PS enhanced DENV membrane fusion and infection [12], it is likely that the supplemented PS reached the LE to mediate DENV membrane fusion. Not surprisingly, when the two enzymes responsible for PS biosynthesis, PTDSS1 AND PTDSS2, were depleted, DENV infection was impaired, furthering the idea that the PS level in the host cell plays an important function in DENV infection. We note that under EMC4 KD, the decrease in the PS level in the LE (Fig 5B) appears more moderate than the block in DENV membrane fusion (Fig 4B) and cytosol escape of the viral genome (Fig 3E). A possible explanation is that there is a strict requirement for the PS level in the LE which is necessary to promote DENV membrane fusion such that any modest decrease in the LE-localized PS level would result in a more pronounced impairment of DENV membrane fusion.

How might EMC4 mediate transfer of PS from the ER to the LE? The EMC was shown to tether the ER to the mitochondria, facilitating PS transfer from the ER to the mitochondria [19]. Moreover, we demonstrated that EMC4, via binding to the LE-associated Rab7 protein, also tethers the ER to the LE [15]. Because the LE-ER MCS is thought to promote PS transport from the ER to the LE [16], we propose that EMC4 might stabilize the LE-ER contact site to facilitate transfer of PS from the ER to the LE. However, because depletion of Rab7 did not block DENV infection, EMC4 must engage a Rab7-independent cellular factor on the LE membrane to support the LE-ER MCS (Fig 5F). We note that a previous study showed that depletion of Rab7 blocked DENV infection [20]. The reason for this discrepancy is unclear, but may be due to the different cell types used between the studies.

What might be an advantage of EMC4-dependent stabilization of the LE-ER MCS that promotes DENV-LE membrane fusion? One possibility is that, in this scenario, release of the viral RNA genome into the host cytosol (which happens after fusion of the DENV and LE membranes) would occur proximal to the ER, where the viral mRNA is required to undergo co-translational translocation. Hence, EMC4 might spatially couple the events of cytosolic escape and protein biosynthesis of the viral RNA genome, circumventing the need to target the viral RNA to the ER after cytosol arrival. Indeed, whether this coupling mechanism operates during entry of other RNA viruses deserves future investigation.

## Materials and methods

### Luciferase reporter virus and replicon system

The luciferase reporter virus luc-DENV2 was generated by overlap extension PCR via fusion of a *Renilla* luciferase (Rluc) with the C-terminal self-cleaving 2A peptide to the DENV-2 strain 16681 capsid in a pD2/IC-30P-NBX background [21]. The luciferase reporter replicon was generated as previously described [22]. Luciferase activity was measured using the Renilla luciferase assay system (Promega, Madison, WI). Luc-DENV2 was used for all infection assays.

### TCID50 for Zika virus (ZIKV) infection

Huh 7.5.1 were infected with 0.1 MOI ZIKV (PRVABC59) in 12-well plates 48 h after siRNA KD. Infected cells were incubated for 3–4 days until widespread CPE was observed in control wells. Culture supernatant was harvested and centrifuged at 3,000 x $g$ for 5 min and transferred to fresh 1.5 mL microcentrifuge tubes through 0.45 micron SFCA syringe filters to remove cell debris. To determine ZIKV titer, 96-well plates were seeded with Huh 7.5.1 at a density 9,000 cells/well and incubated at 37° C and 5% $CO_2$ for 24 h. 10-fold serial dilutions of viral supernatant were plated in a range of $10^{-2}$ to $10^{-7}$ and incubated for 3 days at 37° C and 5% $CO_2$.

To determine TCID50/mL and pfu/mL, cells were fixed in-well with a 1:1 mixture of Methanol and Acetone and incubated at -20° C for 15 min. Excess fixative was aspirated and plates were washed once with 50 mL PBS, then wells were blocked with 25 uL of buffer comprised of 2% BSA and 0.05% Triton X-100 in PBS for 30 min, probed for 50 min with 25 uL of a 1:500 dilution of flavivirus group antibody 4G2 (Genetex GTX57154) in blocking buffer, washed 3 times with 50 uL PBS, and probed for 40 min with 25 uL of a 1:500 dilution of Invitrogen Alexa Fluor 488 Goat anti-Mouse IgG (A11001) in blocking buffer, then again washed 3 times with 50 uL PBS. Wells were then observed under fluorescence using a Nikon Eclipse for presence of flavivirus E-protein. Any well with detectable fluorescence was considered positive for determination of TCID50/mL. TCID50/mL was then determined by Reed-Muench method and pfu/mL was calculated by multiplying TCID50/mL by 0.69.

### DiD labelling of virus

Luc-DENV2 was labelled with a self-quenching concentration of a fluorescent lipid DiD as previously described [12]. Briefly, 10 μl of a 1 mM DiD solution (Thermo Fisher Scientific) was mixed with 1 μl of 10% Pluronic F-127 and bath-sonicated for 5 min. Freshly prepared DiD dispersion was then injected into 1 ml of virus stock (approximately 4 X $10^8$ PFU) under vortexing. The mix was incubated for 30 min at room temperature followed by 2 h at 4°C. Labeled virus was purified from unincorporated dye and from non-viral membranes and proteins by centrifugation (SW55 rotor, 1.5 h, 49,000 rpm, 4°C) on a 40%–25%–20%–15% step gradient of OptiPrep density medium with 0.005% Pluronic F-127. Band between 20% and 25% densities containing DiD-labeled virus was collected. BSA (final concentration 1%) was added to stabilize the preparation. Labeled virus was used within 3 days. Before experiments, the viral suspension was passed through a PES Millipore 0.22 mm filter to remove viral aggregates.

### Cell lines

HEK 293T cells were obtained from ATCC. Huh 7.5.1 cells were a gift from Francis Chisari (Scripps Institute). Cells were cultured at 37°C under 5% $CO_2$ in complete Dulbecco's modified Eagle's medium (DMEM), containing 10% fetal bovine serum (FBS), 10 U/ml penicillin,

and 10 μg/ml streptomycin (Gibco, Waltham, MA). DMEM, Opti-MEM, and 0.25% trypsin-EDTA were purchased from Thermo Fisher Scientific (Waltham, MA). FBS was purchased from R&D systems (Minneapolis, MN).

## siRNA transfection

All Star Negative purchased from Qiagen (Valencia, CA) was used as the control siRNA (labeled as scrambled). Pre-designed siRNAs against different EMC subunits (EMC1 siRNA ID: 122744, EMC2 siRNA ID: 21859, EMC3 siRNA ID: 125192, EMC4 siRNA ID: 140362, EMC5 siRNA ID: 128206, EMC6 siRNA ID: 33278, EMC7 siRNA ID: 28005, EMC8 siRNA ID: 135577, EMC9 siRNA ID: 45418, EMC10 siRNA ID: 41417) and pre-designed siRNA against Rab7 (ID: 139428) were purchased from Thermo Fisher Scientific (Waltham, MA). PTDSS siRNAs were generated through Sigma-Aldrich and target the following sequences: siPTDSS1 sense 5'-GCAGCUGACUGAGUUGAAUTT-3', siPTDSS1 anti-sense 5'-AUU-CAACUCAGUCAGCUGCTT-3'; siPTDSS2 sense 5'-GCACCGAGUCCGAGGUCUATT-3', siPTDSS2 anti-sense 5'-UAGACCUCGGACUCGGUGCTT-3'. Lipofectamine RNAiMAX (Thermo Fisher Scientific) was used as the siRNA transfection reagent. All siRNAs were reverse transfected at the time of cell seeding. Knockdowns were carried out for 48 hours before experiments were conducted.

## Plasmid constructs and transfection

LactC2-GFP plasmid was purchased from Addgene, STARD3-FLAG construct was generated from GFP11-STARD3-FLAG as a template using forward primer 5'- attgttctcgagatgag-caagctgcccagggagctgacccga-3', reverse primer 5'-attgttggatccctacttgtcgtcatcgtctttg-tagtccgcccgggcccccagctcgctgatgcgctgtcgcaggtgaaaggc -3' and pcDNA3 (-) as backbone. 50% confluent Huh 7.5.1 cells were transfected with the indicated plasmids using the FuGENE HD (Promega, Madison, WI) transfection reagent at a ratio of 1:4 (plasmid to transfection reagent; w/v). For HEK 293T cells, polyethylenimine (PEI; Polysciences, Warrington, PA) was used as the transfection reagent. Cells were transfected with the desired DNA construct for at least 24 h before the experiments were conducted.

## Antibodies

Rabbit polyclonal Hsp90 antibody and EMC3 antibody were purchased from Santa Cruz Biotechnology (Santa Cruz, CA), EMC4 antibody from Abcam (Cambridge, MA), rat monoclonal BAP31, rabbit polyclonal EMC5, and EMC7 antibodies from Thermo Fisher Scientific (Waltham, MA), FLAG tag and rabbit polyclonal LAMP1 antibodies from Sigma (St Louis, MO), rabbit monoclonal Rab7 antibody from Cell Signaling Technology (Danvers, MA), rabbit polyclonal EMC6 antibody from Aviva Systems Biology (San Diego, CA), rabbit EMC1 antibody from Abgent (San Diego, CA), actin, rabbit polyclonal EMC2, EMC8, and EMC9 antibodies from Proteintech Group (Chicago, IL), mouse monoclonal J2 antibody from Scicons (Budapest, Hungary), rabbit polyclonal EMC10 antibody from MyBioSource (San Diego, CA), and DENV NS4B antibody from GeneTex (Irvine, CA).

## Quantitative PCR

RNA was isolated from cells using either TRIzol reagent (Thermo Fisher Scientific) or RNeasy kit (Qiagen) followed by cDNA synthesis using iScript reverse transcription supermix for RT-qPCR (BioRad) which served as the template for quantitative PCR (qPCR). qPCR was performed using SsoAdvanced universal SYBR green supermix (BioRad) in CFX Connect Real-

Time PCR Detection System (BioRad). The list of primers used for the qPCR study is shown in Table 1.

## Subcellular fractionation of cytosolic and endosomal fractions

Huh 7.5.1 cells were washed with PBS and semi-permeabilized with 0.05% digitonin followed by centrifugation at 13,000 rpm for 10 min at 4°C. The supernatant was collected and centrifuged again for 30 min at 100,000 rpm to generate a pellet (endosomal) fraction and a supernatant (cytosolic) fraction. Effective separation of the cytosolic and endosomal fractions was confirmed by Western blotting with antibodies against Rab7 (endosome marker), LAMP1 (late endosome/lysosome marker), actin (cytosol marker), and BAP31 (ER marker).

## Virus internalization assay

siRNA-treated Huh 7.5.1 cells were infected with DENV for 1 h at either 37°C or 4°C followed by PBS wash and proteinase K (Roche) treatment for 45 min to remove membrane-associated virus. After proteinase K treatment and PBS wash, total RNA was extracted from the cells and presence of the viral RNA was monitored by RT-qPCR.

## Epifluorescence widefield microscopy and structured illumination microscopy (SIM)

Huh 7.5.1 cells were grown in 12-well plate followed by transfection with specific plasmids with FuGene (Promega) for 24 h where applicable. For knockdown studies, cells were reverse transfected with the desired siRNA using Lipofectamine RNAiMAX (Invitrogen) at the time of cell seeding followed by infection for 48 h where applicable. Cells washed with PBS followed by fixation with 4% formaldehyde at room temperature were then permeabilized using 0.2% Triton X-100, and blocked by 5% milk with 0.2% Tween. Primary antibodies were incubated overnight at 4°C, followed by incubation of fluorescent-conjugated secondary antibodies for 2 h at room temperature. Coverslips were mounted with ProLong Gold mounting medium (Thermo Fisher Scientific) for epifluorescence widefield microscopy. For SIM studies, coverslips were mounted with non-hardening Vectashield antifade mounting medium (Vector Laboratories, CA). Images were taken using inverted epifluorescence microscope (Nikon Eclipse TE2000-E) equipped with 100X oil immersion objective (N.A. 1.4), Sola lumencore light engine and Photometrics CoolSnap HQ camera; or Nikon N-SIM E in 3D-SIM mode with CFI SR HP Apochromat TIRF 100XC Oil immersion objective (NA 1.49), LU-NV series laser unit and ORCA-Flash 4.0 sCMOS camera (Hamamatsu Photonics K.K.). NIS-Elements C software was used to take confocal images and NIS-Elements AR software was used to take SIM images. For 561 channel, illumination modulation contrast was set to 1 and high resolution

**Table 1. Primer sequences used in this study.**

| Primer name | Sequence (5'-3') |
| --- | --- |
| DENV2 RTF | GAGAGACGCTTGGAGAGAAATG |
| DENV2 RTR | CTGAGGGCATGTATGGGTTAAG |
| PTDSS1qRTF | GGCAGCTGACTGAGTTGAATA |
| PTDSS1qRTR | GCTGTGATGCCACCAATAAAG |
| PTDSS2qRTF | CATCTACGACCCAGACAATGAG |
| PTDSS2qRTR | CTTCAGGTACCAGCCAAGAAA |
| GAPDH qRTF | CCCTTCATTGACCTCAACTACA |
| GAPDH qRTR | ATGACAAGCTTCCCGTTCTC |

noise suppression was set to 0.7 while for 488 channel, both illumination modulation contrast and high resolution noise suppression was set to 1 during N-SIM stack reconstruction. FIJI distribution of ImageJ [23,24] was used for image processing, analyses, and assembly. FIJI Coloc2 plugin was used with Costes threshold regression to measure Mander's overlap coefficient and FIJI colocalization threshold plugin was used to generate scatter plot and co-localization pixel map. The analysis of reconstructed SIM images was done on a single z-plane and at least five FLAG-expressing cells were analyzed for every condition during each of three independent experiments.

### Quantification and statistical analysis

All data obtained from at least three independent experiments (biological replicates) were combined for statistical analyses. For Fig 1E, the average of two technical repeats for each of two biological replicates was used for statistical analysis. Results were analyzed using Student two-tailed $t$ test. Data are represented as the mean values and error bar represents standard deviation (SD). $p < 0.05$ was considered to be significant.

### Supporting information

**S1 Fig. Protein levels of each EMC subunit after knockdown (related to Fig 1).** Extracts derived from HEK 293T cells transfected with the indicated siRNA were subjected to SDS-PAGE followed by immunoblotting using the indicated antibodies.
(TIF)

**S2 Fig. Rab7 does not play a role during DENV infection (related to Fig 5).** A. Huh 7.5.1 cells were transfected with the indicated siRNAs and lysed with 1% Triton X-100, and the resulting extract was subjected to SDS-PAGE and immunoblotted with the indicated antibodies. B. Huh 7.5.1 cells were transfected with the indicated siRNAs followed by 48 h infection with luc-DENV2 (MOI 0.05). The data show the relative luciferase unit normalized to the scrambled siRNA, and represent means and standard deviations (SD) ($n \geq 3$). C. Huh 7.5.1 cells were transfected with the indicated plasmid followed by 48 h infection with luc-DENV2 (MOI 0.05). The data show the relative luciferase unit normalized to the WT Rab7, and represent means and standard deviations (SD) ($n \geq 3$). ***$P \leq 0.001$.
(TIF)

### Author Contributions

**Conceptualization:** Parikshit Bagchi, Andrew W. Tai, Billy Tsai.

**Data curation:** Parikshit Bagchi, Kaitlyn Speckhart, Andrew Kennedy, Billy Tsai.

**Formal analysis:** Parikshit Bagchi, Kaitlyn Speckhart.

**Funding acquisition:** Kaitlyn Speckhart, Andrew W. Tai, Billy Tsai.

**Investigation:** Parikshit Bagchi, Kaitlyn Speckhart, Andrew Kennedy, Billy Tsai.

**Methodology:** Parikshit Bagchi, Andrew Kennedy, Billy Tsai.

**Project administration:** Andrew W. Tai, Billy Tsai.

**Resources:** Andrew W. Tai, Billy Tsai.

**Supervision:** Andrew W. Tai, Billy Tsai.

**Validation:** Parikshit Bagchi, Kaitlyn Speckhart, Andrew Kennedy.

**Writing – original draft:** Parikshit Bagchi, Billy Tsai.

**Writing – review & editing:** Parikshit Bagchi, Kaitlyn Speckhart, Andrew Kennedy, Andrew W. Tai, Billy Tsai.

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
