## [Decision Letter · Decision Letter 0]

8 Mar 2022

Dear Prof. Tsai,

Thank you very much for submitting your manuscript "A specific EMC subunit supports Dengue virus infection by promoting virus membrane fusion essential for cytosolic genome delivery" for consideration at PLOS Pathogens. As with all papers reviewed by the journal, your manuscript was reviewed by members of the editorial board and by several independent reviewers. In light of the reviews (below this email), we would like to invite the resubmission of a significantly-revised version that takes into account the reviewers' comments.

Although each of the reviewers found the manuscript of interest and potentially important results, each had some major concerns. Although you will read each one, they consistently conclude that the experimental results are not yet sufficient to justify the conclusions of your manuscript. Each asks for additional experiments which I think will improve the manuscript and convince the reviewers and myself of the conclusions you draw.

We cannot make any decision about publication until we have seen the revised manuscript and your response to the reviewers' comments. Your revised manuscript is also likely to be sent to reviewers for further evaluation.

Sincerely,

Richard J. Kuhn, PhD

Associate Editor

PLOS Pathogens

Sonja Best

Section Editor

PLOS Pathogens

Kasturi Haldar

Editor-in-Chief

PLOS Pathogens

orcid.org/0000-0001-5065-158X

Michael Malim

Editor-in-Chief

PLOS Pathogens

orcid.org/0000-0002-7699-2064

Although each of the reviewers found the manuscript of interest and potentially important results, each had some major concerns. Although you will read each one, they consistently conclude that the experimental results are not yet sufficient to justify the conclusions of your manuscript. Each asks for additional experiments which I think will improve the manuscript and convince the reviewers and myself of the conclusions you draw.

Reviewer's Responses to Questions

**Part I - Summary**

Reviewer #1: This manuscript by Bagchi et al. describes studies of the infection process of dengue virus vs. the role of the ER membrane complex (EMC). Using siRNA depletion the authors find that the critical component of EMC for early steps in DENV infection is EMC4, whose depletion inhibits infection, while single depletion of the other EMC proteins has no effect. Their experiments indicate that EMC4 is involved in the delivery of phosphatidyl serine (PS) to the late endosome (LE) site of DENV fusion. This appears to be via the dual “bridging” interaction of EMC4 with endosomes and the ER. The effect of EMC4 depletion in blocking DENV infection can be rescued by the addition of PS to the cells. The requirement for acidic lipids in DENV fusion in the late endosome was previously detailed in ref. 8, and thus this paper indicates that EMC4 is important to deliver the critical lipid components to the LE.

While overall the paper has some very interesting and potentially important findings, there are a number of experimental and presentation points that need to be carefully addressed.

Reviewer #2: In this manuscript, the authors show the ER membrane complex (EMC) 4 subunit is required for an early stage of DENV infection. Further experiments indicate not entry is blocked post-internalization, but prior to cytosolic RNA release, suggesting that endosomal fusion is blocked. EMC4 depletion also resulted in less late endosomal phosphatidylserine, which has been implicated in DENV fusion. In general, this is an interesting well-done study. The major limitation is that the 2 experimental observations of EMC4 depletion (blocked fusion and less LE PS) have not been experimentally linked.

Reviewer #3: Several previous studies identified the ER membrane protein complex (EMC), a 10-subunit complex embedded in the ER membrane, as a critical host factor for the replication of dengue and other flaviviruses. Subsequent studies elucidated its molecular function for the biogenesis of the viral non-structural proteins NS4A/NS4B, two transmembrane proteins, consistent with its described cellular role in transmembrane protein biogenesis. However, other studies suggested an additional role in facilitating entry of dengue viral particles into host cells, without providing extensive evidence. Bagchi et al. are trying to resolve this by performing siRNA knockdown (KD) studies, cellular fractionation and imaging of fluorescently labelled viral particles. They confirm that viral entry is indeed affected by EMC4 KD using cellular fractionation and DiD-labelled virions. They also provide modest evidence that EMC4 activity is linked to phosphatidylserine transfer from ER to endosomes. However, their claim that viral entry is specifically mediated by a single EMC subunit, EMC4, is not supported sufficiently and could be due to flaws in their experimental setups. The manuscript overall addresses an important question whether an ER-resident host factor can facilitate viral entry directly (in addition to its known function of viral protein biogenesis) but, in its current form, lacks the necessary evidence to have confidence in their conclusions.

**Part II – Major Issues: Key Experiments Required for Acceptance**

Reviewer #1: Major points:

1. The reversal of the inhibition by EMC4 depletion using PS is a critical experiment. However, it is odd that this is the lipid that is used—the original ref 8 report proposed the LE lipid BMP as the key lipid, although both PS and PG could enhance DENV fusion with liposomes or with the plasma membrane. What is known about the lipid specificity that is needed for this effect in their system? Can BMP or PG substitute? How specific is the detection with lact-C2-GFP for PS vs. other acidic lipids? Given that this fluorescent probe shows considerable staining of the LE in the EMC4 depleted cells (Fig. 5A), it is important to address whether the depletion of EMC4 is actually acting via another acidic lipid. While the inhibition of PS-synthesis enzymes is helpful, further experiments that address the lipid specificity are important as well.

2. It is crucial to prove that EMC4 depletion does not inhibit DENV binding or endocytic uptake but rather DENV fusion and RNA release. The evidence for the lack of effect on binding/endocytosis is Fig. 3F. In order to be convincing, the linearity of this RT-qPCR assay vs virus amount bound and/or endocytosed should be included. This would also strengthen the assays in Fig. 3C and 3E. More experimental details should be included here (see point 4 below).

3. The authors find that siRNA depletion of rab7 does not reduce DENV infection (Fig. S2). This is taken as evidence that EMC4 does not require the previously observed binding to rab7 for its action in bridging LE-ER and transferring PS. The lack of effect on infection is puzzling since numerous studies have shown that DENV fuses in the LE, and delivery to this compartment requires rab7. The authors cite one paper in which rab7 depletion by siRNA inhibits DENV infection and comment that they don’t know the reason for the discrepancy. They do not mention the many other studies that found inhibition of DENV infection by dominant-negative rab7. The siRNA depletion looks convincing, but the result here would be strengthened by demonstrating that the depletion is sufficient to block the EMC4-rab7 interaction or downstream effects of that. What would happen with a DN-rab7? As it stands now, this result is not in sufficient depth to be convincing.

4. Overall the paper is quite light on experimental details, which should be added to the legends and/or methods. For example, please indicate the times of siRNA transfection, times of infection, etc. In Fig 3D, please indicate what proportion of the total was loaded vs. the cytosol and endosome fractions. The use of rab7 as a LE marker is odd, since it cycles on and off the membrane/cytoplasm, and a LE protein like CD63 or LAMP1/2 would be a better choice.

5. The end of the discussion is overstated. There is no evidence that the release of the vRNA from the LE at a site close to the ER would promote infection, although it could do so. Conversely though, fusion of the virus with the plasma membrane or transfection of cells with the vRNA is known to efficiently cause infection. The authors should tone down this argument.

Reviewer #2: . One experiment is required to bring this study full circle. In ref. 8 in your manuscript, the authors could enhance fusion with exogenous PS and overcome the fusion defect in the absence of PS with chlorpromazine (CPZ), which resolves hemifusion intermediates. This should be tested in the context of EMC4 depletion. The PS addition would test whether EMC4 is required for LE PS, while complementation of DENV fusion/entry/replication by CPZ would link the two phenotypes. Some variation on this type of complementation experiment would greatly improve the manuscript.

Reviewer #3: 1) Fig. 1A/S1: The authors chose to use siRNA KD of EMC subunits, which is hampered by variability of KD efficiency. Based on this experimental setup they conclude that EMC4 but not other subunits promotes dengue infection as measured using a Luciferase virus at 48h. This is contradictory to numerous other studies (e.g. Savidis et al. ( PMID: 27342126), Ngo et al. (PMID: 31516121)), which showed that all (tested) EMC subunits are critical for flavivirus infection. While the authors show that EMC4 siRNA treatment efficiently reduced EMC4 mRNA levels (~>95% reduction) as well as led to depletion of EMC4 protein, they did not show that other EMC subunits were sufficiently depleted in their experiment. In fact, mRNA levels were only reduced by 70-90%, which may not be sufficient to disrupt EMC function and could explain the lack of phenotype. Savidis et al showed a reduction of DENV replication for siRNAs against multiple EMC subunits, further supporting that the observation here is due to technical reasons. The authors should address this by either using more efficient siRNAs against other subunits and validation of protein depletion by Western Blot, or CRISPR KO/I methods.

2) The authors show that EMC4 siRNA selectively depletes EMC4 protein levels without affecting the expression of other subunits (Fig 1D). This is consistent with a previous report (PMID: 29809151) that showed that EMC4 CRISPRi did not reduce expression levels of other subunits, while EMC2 CRISPRi did. EMC4 may thus not be important for the overall structural integrity of the complex but no conclusion about the functionality of the complex can be drawn. Indeed, genetic interaction mapping and whole-proteome analysis of EMC2 and EMC4 CRISPRi KD experiments suggests that their depletion has similar molecular consequences.

Furthermore, the conclusion that EMC4 does not promote biogenesis of flaviviral NS proteins is not well supported. In Fig 2A the authors used expression of NS4B as a readout whether EMC4 is required for NS4B biogenesis. However, the Ngo et al study (PMID: 31516121) demonstrated that NS4B expression is impaired only in the context of the NS4A-NS4B polyprotein sequence (containing the ER-targeting 2K signal peptide). Additionally, that study used EMC4 KO cells and demonstrated a defect in viral protein biogenesis. The authors should thus repeat the experiment with an appropriate polyprotein construct. Additionally, to make EMC4-specific claims about a lack of effect on polyprotein biogenesis, the authors should also include validated KD of other EMC subunits for comparison.

3) Fig 2C-E: The authors should provide experimental details (timepoint of harvest after siRNA transfection) and control data to ensure that EMC4 protein levels were fully depleted. As the experimental system is a stable replicon cell line, additional controls (e.g. a replication inhibitor or an siRNA targeting a host factor known to affect viral genome replication (e.g. oligosaccharyl transferase) need to be included. Otherwise, it is possible that the observed lack of decrease of DENV RNA levels upon EMC4 treatment is due to the experimental setup, such as timepoint and readout (remaining EMC4 protein needs to be depleted, then viral RNA needs to be degraded. Even then qPCR is generally based on the amplification of short RNA fragments and may obscure an effect on the already present viral RNA). An additional readout on viral protein level should be performed.

4) Fig 3: The experimental setup is not suitable to draw conclusions about early steps in dengue infection. Why do the authors think that differences in viral RNA measured at 48hpi are due to an entry defect? At this timepoint the virus has undergone multiple rounds of entry, translation, genome replication, assembly and egress. The conclusion seems to be based on comparison with the “negative” result from Fig 2E. To measure effects specifically on entry, the authors should collect cells within ~12 hpi, where no/little replication has occurred (ideally in presence of a polymerase inhibitor).

5) As the authors make the claim that EMC4 supports DENV genome release into the cytosol, they should include KD/KO of another EMC subunit for comparison in the cellular fractionation and DiD labelling experiments. This would probe whether indeed EMC4 specifically or rather the entire complex is required for viral entry.

6) Fig 5: The authors use EMC3 KD as a negative control based on the observation in Fig 1 that EMC3 is not important for dengue infection. However, this is contradictory to numerous other studies and may be due to insufficient KD. As mentioned above, KD should be validated on protein level or complete KO cells should be used.

To further support the role of PS in viral entry, the authors could utilize the fractionation assay and DiD assay.

7) The authors should confirm EMC4 and PS dependency for viral entry using another flavivirus (e.g. Zika), for which EMC subunits were also identified as critical host factors. Additionally, the authors should include a negative control virus, which does not utilize late endosomes/EMC.

**Part III – Minor Issues: Editorial and Data Presentation Modifications**

Reviewer #1: Minor points:

1. On P17 of discussion and again at end of discussion please reword. The vRNA is not co-translationally translocated, the nascent polyprotein is.

2. The model in Fig. 5F needs work. It should clearly show that the virus NC assembles on the cytoplasmic face of the ER and buds into the ER lumen, followed by transport of the virus particle in secretory vesicles.

Reviewer #2: 1. The authors normalize their control in all experiments to 100 without any error. This is problematic because it skews p values to be more significant than they are. The error in control samples should be maintained (include error bars) and p-values recalculated. P-values should then be included in figures.

2. In Fig. 2C & 3A, the dsRNA IF is difficult to appreciate. Could you use a higher magnification (5-10 cells visible)?

Reviewer #3: 1) The authors wrote in the introduction: “To identify host factors that facilitate DENV infection, several groups used the genetic CRISPR/Cas9 loss-of-function approach….”

Please provide references to these studies (Zhang et al; Marceau et al; Ma et al; Savidis et al; Hoffmann et al). The statement may also be altered to include other flaviviruses that – like dengue – depend on EMC.

2) Can the authors provide more information on the quantification of the SIM experiment (how many FOVs/cells were quantified?)

3) As PTDSS1/2 are potential flavivirus host factors working in concert with EMC4, can the authors provide information how these genes scored in the CRISPR screens relative to EMC4? Is there activity (largely) redundant, which may preclude enrichment in a single gene KO CRISPR screen?

PLOS authors have the option to publish the peer review history of their article (what does this mean?). If published, this will include your full peer review and any attached files.

Reviewer #1: No

Reviewer #2: No

Reviewer #3: No
---

## [Editor Report · Decision Letter 1]

30 Jun 2022

Dear Prof. Tsai,

We are pleased to inform you that your manuscript 'A specific EMC subunit supports Dengue virus infection by promoting virus membrane fusion essential for cytosolic genome delivery' has been provisionally accepted for publication in PLOS Pathogens.

Best regards,

Richard J. Kuhn, PhD

Associate Editor

PLOS Pathogens

Sonja Best

Section Editor

PLOS Pathogens

Kasturi Haldar

Editor-in-Chief

PLOS Pathogens

orcid.org/0000-0001-5065-158X

Michael Malim

Editor-in-Chief

PLOS Pathogens

orcid.org/0000-0002-7699-2064

This revised version is responsive to all of the reviewers' major and minor comments and suggestions.
---

## [Editor Report · Acceptance letter]

10 Jul 2022

Dear Prof. Tsai,

We are delighted to inform you that your manuscript, "A specific EMC subunit supports Dengue virus infection by promoting virus membrane fusion essential for cytosolic genome delivery," has been formally accepted for publication in PLOS Pathogens.

Best regards,

Kasturi Haldar

Editor-in-Chief

PLOS Pathogens

orcid.org/0000-0001-5065-158X

Michael Malim

Editor-in-Chief

PLOS Pathogens

orcid.org/0000-0002-7699-2064